# Periodontitis Frequently Exists in Patients with Colorectal Carcinoma and Causes Supplementary Impairment of Insulin Resistance

**DOI:** 10.3390/metabo15060414

**Published:** 2025-06-19

**Authors:** Mario Zivkovic, Marko Boban, Domagoj Vrazic, Ivan Vulic, Ivan Budimir, Nina Blazevic, Aleksandar Vcev, Marko Nikolic

**Affiliations:** 1Internal Medicine Clinic, Department of Gastroenterology, University Hospital “ Sestre Milosrdnice”, Vinogradska Cesta 29, 10000 Zagreb, Croatia; ivan.budimir.zg@gmail.com (I.B.); nina.blazevic05@gmail.com (N.B.); marko.nikolic72@gmail.com (M.N.); 2Faculty of Dental Medicine and Health Osijek, University JJ Strossmayer Osijek, Crkvena 21, 31000 Osijek, Croatia; marcoboban@yahoo.com (M.B.); aleksandar.vcev@fdmz.hr (A.V.); 3Department of Cardiology, University Clinic “Bonto”, Radnicka Cesta 1A, 10000 Zagreb, Croatia; 4School of Dental Medicine, University of Zagreb, Gunduliceva 5, 10000 Zagreb, Croatia; vrazic@sfzg.hr; 5Department of Internal Medicine, General Hospital Pula, Santoriova Ulica 24a, 52100 Pula, Croatia; ivan.vulic22@gmail.com

**Keywords:** colorectal carcinoma, periodontitis, diabetes mellitus, glucose intolerance, insulin resistance, HOMA

## Abstract

**Background:** There are known associations between periodontitis and colorectal cancer, but knowledge on the connections existing between the two are not fully understood. The aim of our study was to assess prevalence and clinical severity of periodontitis in patients with localized colorectal cancer. Secondly, the dynamics of metabolic derangements, particularly glucose metabolism, insulin resistance, and diabetes mellitus were studied as well. **Methods:** Diagnostic procedures included endoscopies with patohistology, laboratory exams, the insulin resistance homeostatic model assessment method (HOMA index), anthropometrics, and radiology imaging. Periodontal status was evaluated by full-mouth periodontal examination. **Results:** A total of 79 patients with localized colorectal carcinoma, with slight male predominance of 55.7%, and an age of 65.7 ± 12.4 years participated in this study. Three-quarters of patients (73.4%; 95% CI: 63.0–82.2%) were diagnosed with periodontitis. Patients with periodontitis and colorectal cancer had significantly increased glucose levels (fasting and after oral glucose challenge), (both *p* < 0.05). Also, increased values of the HOMA index were found in patients with periodontitis vs. controls (without periodontitis) and colorectal cancer; 6.38 ± 5.74 vs. 3.58 ± 2.6 (*p* = 0.012); Spearman’s Rho correlation coefficient = 0.271 (*p* = 0.039). There were significant differences in high-density cholesterol in patients with periodontitis vs. controls, 1.41 ± 0.28 vs. 1.23 ± 0.35 mmol/L (*p* = 0.016), but correlations were insignificant; Rho c.c. = 0.045 (*p* = 0.738). **Conclusions:** The most consorted connections between periodontitis and colorectal cancer were found among parameters of metabolic domain, especially glucose concentrations and insulin resistance. Further studies, which would include novel and emerging antidiabetic treatments and their effects on the prevention or control of both diseases, would be warranted.

## 1. Introduction

Colorectal carcinoma is among the most important global public health concerns [1]. There are nearly 2 million new cases a year, making it the third most common cancer globally and the second leading cause of cancer death in the world [2]. The disease is of rising prevalence, connected with the aging population which is burdened with metabolic disorders, a rich caloric diet, a sedentary lifestyle, and obesity [3,4].

High caloric dietary intake, greater prevalence of unsaturated fat, and high sugar are known to be connected with the development of colorectal carcinoma. A similar trend was found among numerous metabolic disorders such as glucose metabolism or diabetes and obesity [5,6]. Prolonged periods of increased glucose concentrations were found to be associated with an overall increase in insulin and insulin-like growth factor-1, the development of insulin resistance, and an increase in fat-derived adiponectin [7,8]. Obesity was associated with an elevation in glucose and insulin concentrations and diminishes adiponectin sensitivity via decreased expression of mRNA for AdipoR1/R2 receptors and leads to glucose intolerance [9,10]. This is a particular background process acting in circumstances of dysregulated energy homeostasis, with increased supplies as in diabetes or metabolic syndrome [10].

Among other functions, it is worth pointing out adiponectin-mediated, i.e., obesity-associated, support of chronic inflammation increases concentrations of tumor necrosis factor alpha (TNF-a) and the interleukin family (IL-6 and IL-18) [9,10,11]. Obesity is associated with a chronic low-grade pro-inflammatory state, where the increased levels of TNF-a, among others, suppress the transcription and secretion of adiponectin [12].

An overall systematic disbalance of regulatory processes, which include the excess production of free radicals, cytokines, and adipose tissue hormones and the continuous activation of low-grade inflammatory processes, supplemented with the impairment of immune reactions and alternations of the gut microbiome in a complexly interconnected and perplexed manner, shares common ground with the development of both metabolic and cancerous disorders [13,14,15].

Even in remote and anatomically unrelated regions, such as the oral cavity, periodontitis was found to be frequently existent and associated with colorectal carcinoma [16]. However, the connections existing between colorectal carcinoma and periodontitis are not fully understood and have not been systematically studied thus far.

Periodontitis is the most frequent chronic inflammatory mouth disease, found in 15% of the adult population [17]. The inflammatory process includes increased production of reactive oxygen radicals and oxidative stress via interaction with acute phase cytokines [17]. The prevalence of periodontitis increases with age, cigarette abuse, and poor oral hygiene [18]. Poor glycemic control was also found to be connected with oral inflammation, even in populations of young children [19]. There are also complex connections with immunodeficiencies, osteoporosis, and various infection agents of oral microbiota that are being investigated [20].

The aim of our study was to assess the prevalence and clinical severity of periodontitis in patients with localized colorectal carcinoma. In addition, this study systematically analyzed the effects of the coexistence of both disorders on metabolic profiles and insulin resistance. Our null hypothesis assumed that there are no connections between insulin residence and periodontists prevalence in patients with colorectal carcinoma.

## 2. Materials and Methods

### 2.1. Study Design

This was a prospective cross-sectional study.

### 2.2. Study Sample

Our study enrolled a consecutive sample of patients who were 18 years and older, diagnosed with nonmetastatic colorectal cancer in tertiary medical at the Center-University Hospital Centre “Sestre milosrdnice” in Zagreb during the period from May 2021 to May 2023. It included patients with pathohistological proof of colorectal adenocarcinoma on endoscopy. Disease staging was assessed by contrast-enhanced multislice computerized tomography. There was no additional follow-up.

Patients with metastatic colorectal cancer, acute intestinal diverticulitis, inflammatory bowel disease, or gluten enteropathy, or who were previously treated for diabetes mellitus or had a history of malignant diseases over the past 5 years were not included in this study. Patients with severe acute illness or acutely decompensated chronic illness were also not included.

### 2.3. Clinical Data and Laboratory Methods

Blood samples were obtained during morning hours 07:30–08:30 a.m. at room temperature (20–22 °C), from the brachial vein, after an overnight fasting. The following parameters were assessed: complete blood count, coagulation, basic biochemistry, hormonal status (vitamin D, ACTH, cortisol, adrenocorticotropic hormone, thyroid stimulating hormone, peripheral thyroid hormones [FT3 and FT4], insulin, insulin-like-growth factor, and human growth hormone), anti tTg, total IgA, tumor markers, and 2 h OGTT (oral glucose tolerance test) with determination of glucose and insulin fasting values and values two hours after a standardized carbohydrate load. The HOMA index of insulin resistance was calculated over the concentrations of fasting glucose and fasting insulin [21].

The clinical data were obtained by two gastroenterology specialists, and included medical history, diagnostic exams, a patient interview, and a specific factor assessment, which included age, gender, and smoking.

The body composition of patients was analyzed using the multifrequency bioelectrical impedance method (TANITA SC scale −240). The analysis of body composition which included metabolic years, body mass (kg), body mass index BMI (kg/m^2^), adipose tissue (kg), visceral fat (level), muscle mass—total (kg), skeletal muscle mass (kg), and bone mass (kg) was carried out in an upright standing position on the specified scale by two trained nurse.

### 2.4. Periodontology Profile

The patients were examined by two experienced periodontology specialists. All clinical measurements were performed with a standard PCP 15 periodontal probe (UNC 15, Hu Friedy^®^, Chicago, IL, USA). Clinical assessment parameters included the following: full mouth bleeding score (FMBS), probing depth (PD), gingival recession (GR), and attachment level (CAL). Probing depth (PD) was measured as the distance between the gingival margin and the deepest aspect of the periodontal pocket. Gingival recession was assessed as the distance from the enamel–cement junction to the marginal edge of the gingiva, and the attachment level (CAL) was assessed as the distance between the enamel–cement junction of the tooth and the deepest aspect of the periodontal pocket. All periodontal examinations were performed on 6 surfaces per tooth. Diagnosis and staging of periodontitis were defined according to the classification of periodontal diseases and conditions from 2017 [22].

### 2.5. Ethical Approval

This study adhered to the Declaration of Helsinki and good clinical practice guidelines. Patients signed informed consent before inclusion and data acquisition. There was no funding, compensation, or other sources of financing. There were no additional reimbursements or in-kind compensation. This study was approved by the Ethical Committee of the Sisters of Charity University Hospital (No 003-06/20-03-015) and the Ethical Committee of the School of Dental Medicine (No 05-PA-30-XVII-5/2020), University of Zagreb.

### 2.6. Statistical Analyses

Power analyses of sample sizes were performed using t-test power analyses (MedCalc statistical software 19.6) for independent groups of population with periodontitis vs. no-periodontitis, with expected higher values of the Homa index of 75% in the group with periodontitis (7.0 ±5.0, compared to 4.0 ±3.0), alpha < 0.05, and test power of 80%. At least 39 subjects needed to be included in the group with periodontitis, while 20 subjects needed to be included in the control group (respecting the group size of 2:1 in favor of the group with periodontitis).

Data analyses were performed by an experienced statistician using IBM SPSS for Windows software, version 29.0.1. The Kolmogorov–Smirnov test was executed, and nonparametric tests and data presentation were used in subsequent proceedings. Continuous data were presented as averages with standard deviations or medians and interquartile ranges. Differences between groups (independent differences) were analyzed with the Mann–Whitney U test. Correlations between the clinical stage of periodontitis and clinical data parameters were assessed by Spearman’s rank correlation. *p* values less than 0.05 were considered statistically significant. Data were presented in tables and figures.

## 3. Results

### 3.1. Patients Sample

There were 79 patients with localized colorectal carcinoma, with a slight male predominance at 55.7% (N = 44). The mean age was 65.7, in the range 33–84, and patients with periodontitis were older (58.71 ± 14.07 vs. 68.22 ± 10.76, *p* = 0.006). Nearly one-quarter of patients were active nicotine consumers (26.6%). A total of 43 patients (54.4%) were diagnosed with colon cancer, and the remaining 36 had rectal cancer (45.6%).

Three-quarters of patients (73.4%; 95% CI: 63.0–82.2%) were diagnosed with periodontitis. Twenty-five female patients had periodontitis and colorectal cancer, whilst ten had only colorectal cancer. The most common type of periodontitis in our sample of patients with nonmetastatic colorectal carcinoma was the most severe, i.e., grade IV (n = 31; 53.4%), followed by grade III (n = 21; 36.2%), whilst there were only 4 patients with grade I and 2 patients with grade II.

### 3.2. Metabolic Profile

A notable number of patients were diagnosed with diabetes mellitus (n = 30; 38%) or glucose intolerance (n = 18; 22.8%). The average body weight was 81.5 kg, in the range 49.2–135.3 kg; the average body mass index was 27.75 kg/m^2^, in the range 17.9–43 kg/m^2^. Body composition included an average muscle mass of 53.31 kg, in the range 32.0–78.3 kg, whilst the average fat mass was 25.38 kg, in the range 6.6–66.3 kg, and the average bone mass was 2.81 kg, in the range 1.7–4 kg.

There were 34 (43.0%) overweight patients with a body mass index (BMI) of 25–29.9 kg/m^2^, and 22 (27.8%) were obese, with a BMI over 30 kg/m^2^.

A greater part of the patients with periodontitis, 39 (67.2%) also had impaired glucose tolerance or diabetes, while normal glycemic status was found only in 19 (32.8%) (*p* = 0.049). There were no significant differences considering the existence of periodontitis in the remainder of the studied groups, including for the variables of gender, tumor localization, or cigarette abuse (all *p* > 0.05, respectively).

When body type was taken into account, patients with a BMI > 25 kg/m^2^ (n = 56) had periodontitis in 80.4% of cases (n = 45), whilst patients with a BMI < 25 kg/m^2^ (n = 23) had a significantly lesser prevalence of periodontitis: 56.5% (n = 13) (*p* = 0.029).

Furthermore, the existence of periodontitis was most commonly found in overweight patients (BMI 25–29.9 kg/m^2^) with prevalences of 50% (n = 29) vs. 23.8% (n = 5) [without periodontitis], followed by obese patients (BMI > 30 kg/m^2^): 27.6% (n = 16) vs. 28.6% (n = 6); normal weight patients (BMI 18.5–24.9 kg/m^2^) had a prevalence of periodontitis of 22.4% (n = 13) vs. 47.6% (n = 10), respectively. In the subclass analysis of overweight vs. normal weight patients, excluding the obese patients, there was also a significant difference in the existence of periodontitis: 29 (69.1%) vs. 13 (30.9%), *p* = 0.016, respectively.

Patients with periodontitis had higher metabolic years (55.57 ± 18.36 vs. 63.35 ± 13.44, *p* = 0.044).

Differences and correlations of studied parameters depending on the existence of periodontitis are shown in Table 1.

The dynamics of glucose after challenge on the 2 h oral glucose tolerance test (fasting glucose values—mmol/L 6.06 ± 1.10 vs. 7.31 ± 2.32, *p* = 0.016—and glucose values 2 h after standardized carbohydrate load—mmol/L 8.02 ± 3.48 vs. 10.78 ± 4.28, *p* = 0.005) and differences in the HOMA index (3.58 ± 2.61 vs. 6.38 ± 5.74, *p* = 0.012) depending on the existence of periodontitis are shown in Figure 1 and Figure 2.

Other studied parameters, including PTH, TSH, T3, T4, cortisol, human-grown hormone, platelets count, total proteins, CRP, and fecal calprotectin, were all nonsignificantly different and without correlation depending on the presence of periodontitis.

This section may be divided by subheadings. It should provide a concise and precise description of the experimental results, their interpretation, as well as the experimental conclusions that can be drawn.

## 4. Discussion

This study analyzed the coexistence of colorectal cancer and periodontitis. Specifically, it was the first study which analyzed the combined effects of glucose metabolism disorders. The latter was further analyzed in relation to prevalence of diabetes mellitus or glucose intolerance, in addition to glucose concentrations, insulin resistance, and the homeostatic model assessment (HOMA) index [23].

The solid connection existing between colorectal cancer and periodontitis was reaffirmed in our study as well [24]. An astonishingly high 73.4% of patients with localized colorectal cancer had periodontitis. The most common types of periodontitis were the most severe (grade IV and III) [25]. Our study is congruent with the fact that periodontitis is a risk factor for and a potent promoter of colorectal cancer, especially as the more pronounced clinical severity of oral pathology was, at the same time, the most represented among patients [25,26]. Although the sequential nature and causality of the two diseases remains not entirely understood, there are numerous ongoing research projects, even on the effects of the oral and gut microbiome, where the pathogens from the oral cavity were found to promote the growth of various tumors (oral squamous-cell carcinoma, colorectal cancer, and pancreatic ductal adenocarcinoma), and could be even utilized as a lower gut biomarker [27,28]. Interestingly, we found no differences or correlations for cigarette abuse and the coexistence of the diseases, similar to other studies which in part were independent for smoking history [29].

On the other hand, an overwhelming number of associations were found between the existence of periodontitis and localized colorectal cancer via the changes in metabolism [30]. Nearly three quarters of the studied sample had either been diagnosed with diabetes mellitus or glucose intolerance [16]. Although the associations and potential mechanisms among diabetes and two conditions were not previously analyzed in-depth through the available studies, there are known interconnections within pathways of the cell cycle or chronic low-grade inflammatory processes, particularly the acute phase reactants, such as TNF-a, that were found to cofound the pathway [31]. This was in line with the already well-known fact of long enduring insulin resistance in patients with acute bacterial or viral infections [32]. Similar behavior was also found in our study, but on the grounds of a cancerous cause. We have, for the first time, described that the values of the HOMA index and glucose concentrations, in patients with localized colorectal cancer, were both significantly different and meaningful correlated depending on the existence of periodontitis [33]. Even more important is the fact that both fasting glucose concentrations and the results of the 2 h oral glucose tolerance test of challenged glucose concentrations were both increased in patients with localized colorectal cancer and coexisting periodontitis. A similar process, with increased insulin resistance and glucose and leucine concentrations, was previously described in patients with chronic liver cirrhosis [34]. To the best of our knowledge, there are no available studies on the dynamics of glucose and insulin metabolism in coexisting periodontitis and colorectal cancer. There was a significant relationship of metabolic age with a combination of periodontitis and cancer, which underscores the metabolic relationships found among numerous studied parameters from metabolic and lifestyle domains [35].

In our study, there were no differences in terms of body compartment contents (proportions of fat, muscle, and bone tissue) in patients with periodontitis vs. controls. However, there was a discriminative and discontinuous relationship between body mass index and the coexistence of both disorders [31]. The existence of periodontitis was most commonly found in overweight patients, followed by obese patients, and was the least prevalent in normally weighted individuals. In subclass analysis of overweight vs. normal weight patients, excluding the obese patients, there was a significant difference in the existence of periodontitis: 29 (69.1%) vs. 13 (30.9%), *p* = 0.016. This generally explains the fact that there were no significant correlations between BMI and (co) existing periodontitis and nonmetastatic colorectal cancer. Correspondingly, 80.4% of studied patients with a BMI ≥ 25 kg/m^2^ had periodontitis and colorectal cancer vs. 56.5% for those with a BMI < 25 kg/m^2^ [36]. There was a significant correlation between age and the prevalence of periodontitis in patients with colorectal cancer. However, one must note that the relationship is complex and complicated by a number of other factors [37]. There were no significant differences among the other studied parameters, namely PTH, TSH, T3, T4, cortisol, human-grown hormone, platelets count, total proteins, CRP, and fecal calprotectin, depending on the presence of periodontitis. The limitations of our study include the relatively low number of patients and the lack of a long-term prognostic follow-up. There is likewise a limitation in the availability of objectively quantified longitudinal data for lifestyle (particularly the history of diet regimen) and genetic background of patients, which also, in part, narrows the external reproducibility of our study. Although a causative link from the studied settings was not able to be firmly defined, numerous changes in the metabolic domain remain grounds for future studies.

## 5. Conclusions

There are numerous complicated mechanisms in the relationship between periodontitis and colorectal cancer. The most resourceful connections were found among diagnostic parameters in the metabolic domain, which considers alternations in glucose concentrations and insulin resistance. This research provides novel insights into the interrelationship between periodontitis, colorectal cancer, and metabolic dysregulation, with a particular emphasis on insulin resistance. The identification of significantly elevated glucose levels and HOMA index values in patients presenting with both conditions contributes meaningfully to the growing body of evidence on the metabolic crosstalk underlying these comorbidities. It might also increase the awareness of the relevance of diagnostic evaluations for studying the clinical implications on the coexistence of these two seemingly unrelated conditions.

Further studies are warranted to explore the potential role of novel and emerging antidiabetic therapies in the context of systemic disease interactions. In particular, studies that assess the impact of these treatments on the incidence, progression, and clinical outcomes of both periodontitis and colorectal cancer could provide critical insights. Such investigations may help to elucidate whether improved glycemic control and modulation of insulin resistance through these agents can contribute not only to metabolic regulation but also to the prevention and management of associated inflammatory and neoplastic conditions.

## Figures and Tables

**Figure 1 metabolites-15-00414-f001:**
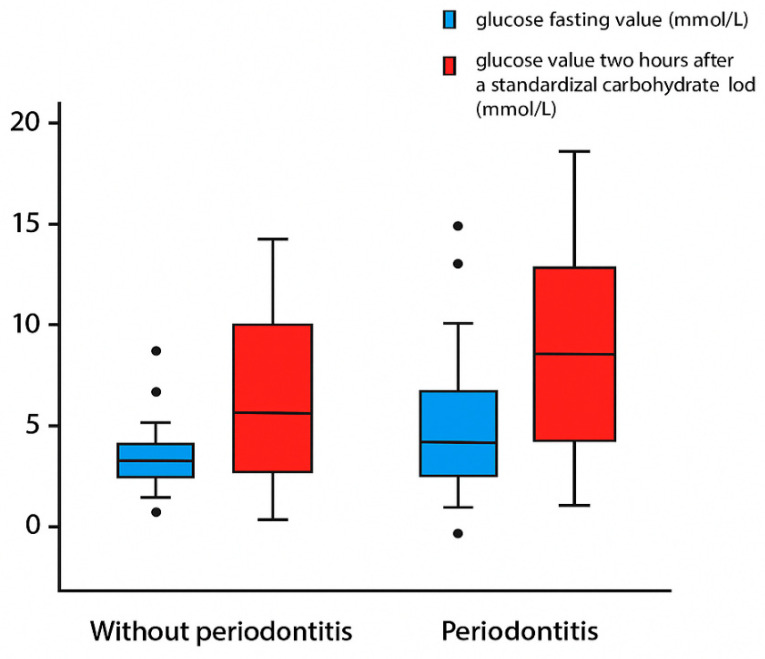
**Fasting and postprandial glucose levels in individuals with and without periodontitis.** Box plots display fasting glucose levels (blue) and glucose levels two hours after a standardized carbohydrate load (red) for individuals without periodontitis (left) and with periodontitis (right). Both fasting and postprandial glucose levels are elevated in the periodontitis group, with wider interquartile ranges and more outliers, indicating greater variability in glycemic control. The differences were statistically significant for both fasting glucose (*p* = 0.016) and postprandial glucose (*p* = 0.005), suggesting a strong association between periodontitis and impaired glucose metabolism.

**Figure 2 metabolites-15-00414-f002:**
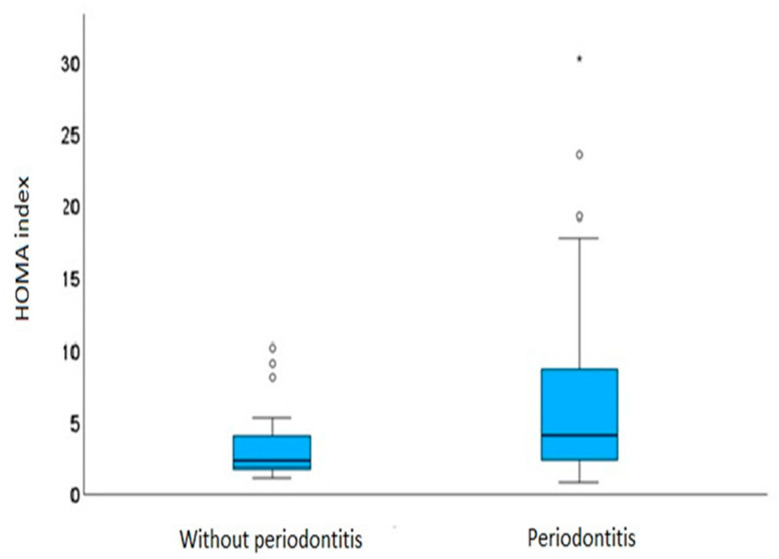
Effect of periodontitis on homeostatic model assessment of insulin resistance (HOMA index). The data illustrate a higher median HOMA index in individuals with periodontitis (right), consistent with the reported *p*-value of 0.012, indicating a statistically significant difference.

**Table 1 metabolites-15-00414-t001:** Dynamics of studied parameters depending on existence of periodontitis.

Mean ± SD	Mann–Whitney		*p* *	Spearman’s Rho (*p*)
	Controls	Periodontitis		
	N = 21	N = 58		
Age (years)	**58.71 ± 14.07**	**68.22 ± 10.76**	**0.006**	**0.299 (0.022)**
Metabolic years	**55.57 ± 18.36**	**63.35 ± 13.44**	**0.044**	0.167 (0.211)
Body weight (kg)	78.79 ± 17.96	82.46 ± 18.56	0.427	−0.144 (0.281)
Body mass index-BMI (kg/m^2^)	26.24 ± 5.41	28.27 ± 4.82	0.111	−0.059 (0.662)
Adipose tissue weight (kg)	23.35 ± 12.16	26.11 ± 10.84	0.297	−0.086 (0.519)
Muscle tissue weight (kg)	52.65 ± 9.79	53.56 ± 12.02	0.894	−0.126 (0.347)
Skeletal muscle tissue weight (kg)	31.29 ± 5.87	31.91 ± 7.14	0.881	−0.126 (0.347)
Bone tissue weight (kg)	2.79 ± 0.49	2.82 ± 0.59	0.885	−0.139 (0.297)
Vitamin D (nmol/L)	47.29 ± 24.26	45.62 ± 23.59	0.824	0.187 (0.159)
Human growth hormone (ng/mL)	1.23 ± 1.78	1.09 ± 1.29	0.786	0.059 (0.662)
IGF-I (ng/mL)	135.95 ± 38.51	128.91 ± 44.39	0.289	−0.093 (0.486)
Hemoglobin (g/L)	127.76 ± 18.23	122.38 ± 22.02	0.444	**−0.296 (0.024)**
Glucose fasting (mmol/L)	**6.06 ± 1.10**	**7.31 ± 2.32**	**0.016**	**0.326 (0.013)**
Glucose 2 h after standardized carbohydrate load (mmol/L)	**8.02 ± 3.48**	**10.78 ± 4.28**	**0.005**	**0.327 (0.012)**
Insulin fasting (mU/L)	12.61 ± 7.01	19.08 ± 17.22	0.053	0.195 (0.142)
Insulin 2 h after standardized carbohydrate load (mU/L)	77.00 ± 65.15	127.05 ± 143.88	0.058	0.161 (0.226)
Homa index	**3.58 ± 2.61**	**6.38 ± 5.74**	**0.012**	**0.271 (0.039)**
Albumin (g/L)	44.73 ± 2.78	43.69 ± 2.00	0.128	−0.118 (0.378)
Cholesterol (mmol/L)	5.34 ± 1.19	4.92 ± 1.06	0.269	−0.021 (0.874)
Triglycerides (mmol/L)	1.23 ± 0.59	1.23 ± 0.41	0.663	−0.007 (0.96)
LDL (mmol/L)	3.25 ± 1.07	3.02 ± 0.87	0.582	−0.032 (0.811)
HDL (mmol/L)	**1.41 ± 0.28**	**1.23 ± 0.35**	**0.016**	0.045 (0.738)
Creatinine (umol/L)	72.05 ± 9.43	78.78 ± 18.46	0.124	−0.031 (0.815)
Urea (mmol/L)	4.96 ± 1.61	5.77 ± 1.93	0.114	**0.369 (0.004)**

Note: *p* = statistical significance; significance number presented in bolded text (*p* < 0.05); BMI = body mass index; N = number of patients; IGF-I = insulin-like growth factor 1; HOMA index = homeostatic model assessment for insulin resistance; LDL = low-density lipoprotein; HDL = high-density lipoprotein; kg = kilograms; kg/m^2^ = kilograms per square meter; nmol/L = nanomoles per liter; ng/mL = nanograms per milliliter; g/L = grams per liter; mmol/L = millimoles per liter; mU/L = milliunits per liter; umol/L = micromole per liter.

## Data Availability

The datasets used and/or analyzed during the current study are available from the corresponding author upon reasonable request.

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
