# Peer review of "Periodontitis Frequently Exists in Patients with Colorectal Carcinoma and Causes Supplementary Impairment of Insulin Resistance"

_metabolites, 2025, doi:10.3390/metabo15060414_

Round 1

Reviewer 1 Report

Comments and Suggestions for Authors

Greetings,

Firstly, I would like to thank the authors for their efforts and work in this article.
In my opinion, the article does not seem to show the relationship of periodontitis and colorectal carcinoma and the writing of the article could be improved.

Reviewer 2 Report

Comments and Suggestions for Authors

The paper “Periodontitis frequently exists in patients with colorectal carcinoma and causes supplementary impairment to the insulin resistance”

conducted by Mario Zivkovic et al. provide very important results regarding the connection between periodontitis and colorectal cancer, with an impact on metabolic parameters, such as insulin resistance.

The main question concerns the coexistence of colorectal cancer and periodontitis. The benefits of the research may be important and create the premises for other studies, as this research confirmed the strong link between colorectal cancer and periodontitis, and further research is necessary in the context of metabolic dynamics. The study may have a relevant impact on future therapeutic protocols used in the treatment of diabetes, which could have beneficial effects in the control of both diseases.

Also, a beneficial impact could be obtained on public health, considering the increased frequency of oral pathology, because it demonstrates that periodontitis may be a risk factor for colorectal cancer. Also, for the first time, the statistically significant correlation between HOMA index values ​​and glucose concentration in patients with colorectal cancer, dependent on the presence of periodontitis, is described.

The formulated conclusions are in accordance with the obtained results, based on the proposed study objectives.

The references are recent and relevant to the information presented.

To improve the quality of the paper, I have the pleasure of suggesting the following to the authors:

  1. Section 2.3. and 3.1. – I would prefer to redefine the titles, to be more suggestive.
  2. Table 1 – needs improvements
  3. Figures – it would be better to be clearer. To represent the parameters on the y-axis (Figure 1).

Reviewer 3 Report

Comments and Suggestions for Authors

The article is interesting. The following may be added to make the purpose and methodology of the study more clear.

  1. What was the rationale to investigate the association of colorectal Ca and periodontitis? Add relevant studies in the introduction.
  2. Explain metabolic years and its relevance.
  3. What was the calculated sample size. The number of control patients is less. How will you define the controls?
  4. Was gingivitis in control patients with colorectal cancer a consequence of poor metabolic control.
  5. What was the stage and grade of periodontitis in these patients? Was there any correlation between the stage of periodontitis and colorectal ca? How many women had periodontitis and colorectal ca.
  6. Many risk factors are shared between the two diseases. What was the effect of confounding factors? Age, high blood glucose, low HDL can independently be associated with periodontitis.
  7. Where are the tables regarding the periodontal assessments. Was there any correlation between the bleeding scores and the BMI, adipose tissue, blood glucose levels, cholesterol and TG levels.
  8. Could long term glucose control (HbA1) better represent the glycemic status, which would have been more relevant in these patients.
  9. HOMA index scores were high in both groups of patients but relatively more in the periodontitis group. Could periodontitis contribute to insulin resistance additionally.
  10. Which of the assessed parameters have a role in prediction of colorectal cancer.
  11. Does periodontitis presence reduce the prognosis of colorectal cancer. What are the implications.
  12. Were any of the patients also having inflammatory bowel diseases.
  13. What is the role of inflammation. Can periodontitis be a source of low grade chronic inflammation?
  14. There are several studies linking periodontal disease to colorectal Ca. The authors can discuss these in the context of the findings of this study.
  15. How many patients were diabetic and on medication. Is there any relevance to the cancer status.
  16. The objectives of the study are not clear. The conclusion can be re-written with relevance to findings from the study.

Reviewer 4 Report

Comments and Suggestions for Authors

Dear Authors,

Thank you for submitting your manuscript to Metabolites. I have thoroughly reviewed your paper and find it to be a valuable contribution to the literature examining the relationships between periodontitis, colorectal cancer, and metabolic dysregulation. The study presents novel findings regarding the association of these conditions with insulin resistance and glucose metabolism.

Your study demonstrates a high prevalence (73.4%) of periodontitis in patients with localized colorectal cancer, with most cases being severe (grades III and IV). The findings regarding glucose metabolism are particularly noteworthy, showing significantly increased fasting glucose levels, post-challenge glucose levels, and HOMA index values in colorectal cancer patients with periodontitis compared to those without.

Abstract

  • The abstract is concise but could benefit from more specific numerical data, particularly regarding the strength of correlations found.

Introduction

  • While comprehensive, the introduction could be more focused on the specific aims of the study. Consider shortening some of the background on adiponectin and refining the research question presentation.

Methods

  • The patient selection criteria are clear, though consider providing more detail on how the sample size was determined beyond the power analysis mention.
  • The bioelectrical impedance methodology section could include information on standardization of measurements (time of day, hydration status).

Results

  • Table 1 is informative but contains some parameters without sufficient explanation in the text (e.g., metabolic years).
  • Figures 1 and 2 effectively illustrate key findings, but would benefit from more detailed captions with statistical significance indicators.

Discussion

  • The discussion effectively addresses the novel findings but could better explore potential mechanistic pathways connecting periodontitis, colorectal cancer, and insulin resistance.
  • The paragraph on fecal calprotectin (lines 320-325) seems disconnected from the main narrative and could be either expanded or removed.

Conclusion

  • The conclusion is appropriate but somewhat brief. Consider elaborating on specific clinical implications and more concrete directions for future research.

Several minor editorial issues require attention:

  1. Line 44: "There are nearly 2 million new cases a year" - Consider updating with the most recent statistics.
  2. Lines 63-64: "...it is worth pointing out adiponectin-mediated, i.e. obesity obesity-associated..." - Remove the repetition of "obesity".
  3. Lines 96-104: Improve formatting of the paragraph on patient selection.
  4. Throughout the paper: Check for consistent hyphenation of terms like "non-metastatic" vs "nonmetastatic".
  5. Ensure consistent spacing after punctuation marks throughout.

This manuscript presents valuable new insights into the relationships between periodontitis, colorectal cancer, and metabolic dysregulation, particularly regarding insulin resistance. The finding that patients with both conditions show significantly higher glucose levels and HOMA index values represents an important contribution to understanding the complex interplay between these diseases.

With the suggested revisions, I believe this paper will make a significant contribution to the literature and be of interest to researchers and clinicians in gastroenterology, periodontology, and metabolic medicine.

Reviewer 5 Report

Comments and Suggestions for Authors

Thank you for the submission of this interesting work.

This manuscript refers to the prevalence and clinical severity of periodontitis in patients with localized colorectal cancer and the dynamics of metabolic derangements, particularly glucose metabolism, insulin resistance and diabetes mellitus.

The topic is within the journal scope, and it can help to understand connections between periodontitis and colorectal cancer and among parameters of metabolic domain, especially glucose concentrations and insulin resistance. This study analyzed coexistence of colorectal cancer and periodontitis. Specifically, it was the first study which analyzed combined effects of glucose metabolism disorders.

In the Materials and Methods section please organize the first two paragraphs into subheadings: Study design and ethical approval, Study sample (place and date of recruitment, inclusion and exclusion criteria, sample size calculation).

In the Discussion section, outline the study’s limitations, as well as practical implications of the study.

Please rewrite Reference section according to instructions for authors:

“1 Dekker E, Tanis PJ, Vleugels JLA, Kasi PM, Wallace MB. Colorectal cancer. Lancet 2019; 394(10207): 1467-1480 [PMID: 31631858 436 DOI: 10.1016/S0140-6736(19)32319-0] 437

2 Baidoun F, Elshiwy K, Elkeraie Y, Merjaneh Z, Khoudari G, Sarmini MT, Gad M, Al-Husseini M, Saad A. Colorectal Cancer 438 Epidemiology: Recent Trends and Impact on Outcomes. Curr Drug Targets 2021; 22(9): 998-1009 [PMID: 33208072 DOI: 439 10.2174/1389450121999201117115717]”

  1. Author 1, A.B.; Author 2, C.D. Title of the article. Abbreviated Journal Name Year, Volume, page range.
  2. Author 1, A.; Author 2, B. Title of the chapter. In Book Title, 2nd ed.; Editor 1, A., Editor 2, B., Eds.; Publisher: Publisher Location, Country, 2007; Volume 3, pp. 154–196.”

In the Reference section 16 out of 38 references was within the last 5 years

Round 2

Reviewer 1 Report

Comments and Suggestions for Authors

Thank you for revising the manuscript.

But in my opinion, the article lacks to show the relationship of periodontitis and colorectal carcinoma

Reviewer 2 Report

Comments and Suggestions for Authors

The changes made have significantly improved the article.
